# Multimodal Latent Representation Learning for Video Moment Retrieval

**DOI:** 10.3390/s25144528

**Published:** 2025-07-21

**Authors:** Jinkwon Hwang, Mingyu Jeon, Junyeong Kim

**Affiliations:** Department of AI, Chung-Ang University, Seoul 06974, Republic of Korea; wlsrnjs905@cau.ac.kr (J.H.); smart2557@cau.ac.kr (M.J.)

**Keywords:** video moment retrieval, visual language reasoning, multimodal representation learning

## Abstract

The rise of artificial intelligence (AI) has revolutionized the processing and analysis of video sensor data, driving advancements in areas such as surveillance, autonomous driving, and personalized content recommendations. However, leveraging video data presents unique challenges, particularly in the time-intensive feature extraction process required for model training. This challenge is intensified in research environments lacking advanced hardware resources like GPUs. We propose a new method called the multimodal latent representation learning framework (MLRL) to address these limitations. MLRL enhances the performance of downstream tasks by conducting additional representation learning on pre-extracted features. By integrating and augmenting multimodal data, our method effectively predicts latent representations, leveraging pre-extracted features to reduce model training time and improve task performance. We validate the efficacy of MLRL on the video moment retrieval task using the QVHighlight dataset, benchmarking against the QD-DETR model. Our results demonstrate significant improvements, highlighting the potential of MLRL to streamline video data processing by leveraging pre-extracted features to bypass the time-consuming extraction process of raw sensor data and enhance model accuracy in various sensor-based applications.

## 1. Introduction

The rise of artificial intelligence (AI) has ushered in transformative changes across various sectors, particularly in tasks involving the analysis and processing of data from visual sensors. The significance of video-based tasks continues to grow, fueled by the increasing demand to harness vast volumes of visual information for applications such as surveillance, autonomous driving, and personalized content recommendation systems. Despite its potential, leveraging video data presents unique challenges, primarily related to the complexities and time-intensive nature of feature extraction.

One of the most prominent limitations in utilizing video data is the extensive duration required for model training, further compounded by the high computational demands. This issue is particularly pressing for research labs that lack access to advanced hardware, such as a GPU, which significantly hinders their ability to conduct comprehensive research. As depicted in Figure 1, the most time-consuming aspect of training models with video data involves extracting features from the original video. Figure 1a illustrates this typical pipeline, where augmentation is applied to the raw data before features are extracted by an encoder for the downstream task. In contrast, our work explores the approach shown in Figure 1b, which simplifies this process by omitting the costly extraction step and operating directly on pre-extracted features. This procedure is especially laborious for long-term videos.

Researchers often resort to using pre-extracted features to address the time constraints associated with feature extraction [1,2,3,4,5]. These pre-extracted features can be employed in downstream tasks such as video moment retrieval (VMR) [6,7,8,9,10]. However, this approach typically involves utilizing the pre-extracted features without further enhancement or adaptation, which may limit the overall performance improvements. This inherent limitation of existing approaches constitutes the core technical gap this paper aims to address.

To address this technical gap, we propose a new method called multimodal latent representation learning (MLRL). This method enhances the performance of downstream tasks by conducting additional representation learning on pre-extracted features. Our work is inspired by recent studies demonstrating the effectiveness of using latent representation prediction in representation learning frameworks [11,12,13,14]. Models in the JEPA family [11,12,14] create a single context block from the original data and use it to predict multiple target blocks. Unlike traditional methods that rely on techniques like contrastive learning [15,16,17,18,19,20] or pixel-level reconstruction [21,22], JEPA models do not require separate augmentation for sample selection or human-annotated data, which simplifies the process and reduces dependency on extensive data augmentation and manual annotations. Inspired by these methods, we propose the MLRL framework, which performs additional representation learning on pre-extracted multimodal data. Our proposed approach involves integrating pre-extracted video clip data and text query data, augmenting these features to ensure diversity, and then dividing the combined features into context and target blocks. The latent representation of the target block is predicted from the context block, facilitating more efficient and effective learning. This method offers several notable advantages: it significantly reduces the time required for feature extraction in downstream tasks and enhances performance compared to traditional approaches that rely solely on downstream task learning. Additionally, the proposed method is versatile and can be applied to various video-related tasks.

We specifically select the video moment retrieval (VMR) task as an ideal testbed because its core characteristics align perfectly with the objectives of our MLRL framework. As an inherently multimodal task, VMR requires a deep understanding of both visual and textual information, allowing us to directly validate our multimodal latent prediction objective. Furthermore, the reliance of many state-of-the-art VMR models on pre-extracted features and the recent emergence of powerful LLM-based approaches in this domain—such as LLaVA-MR [23] and InternVideo2 [24], which have set new state-of-the-art benchmarks—makes it the perfect scenario to demonstrate the value of our enhancement framework. To validate our proposed method, we utilize the QVHighlight dataset [1] for the task of video moment retrieval. This standardized benchmark is well-suited for our evaluation as it provides the necessary pre-extracted features and supports fair, direct comparison with existing works. Our method is benchmarked against the baseline model QD-DETR [2] within this dataset, demonstrating its efficacy. In addition, we evaluate two training regimes—a separate two-stage pipeline and a joint one—because many GPU-limited labs prefer to refine the encoder once offline and later reuse it across multiple retrieval heads without rerunning the costly detector. Contrary to prior VMR approaches that merely fine-tune frozen clip features, the proposed framework iteratively updates a unified clip-query embedding by means of a JEPA-inspired latent-prediction objective. To the best of our knowledge, this synergy between pre-extracted features and on-the-fly multimodal refinement has not yet been explored in the VMR literature. In summary, this paper addresses the challenges of utilizing original video data, introduces a new representation learning method leveraging pre-extracted feature data, and showcases how this approach can achieve superior performance in downstream tasks.

Our contributions are as follows. (1) We propose MLRL, a framework specifically designed to enhance pre-extracted multimodal features from sensor data, addressing the performance gap left by methods that use these features without refinement. (2) We demonstrate the successful application of a JEPA-inspired latent prediction objective to a multimodal (video-text) context, extending its typical single-modality application. (3) Our framework achieves significant performance gains on the QVHighlight benchmark, improving upon a strong QD-DETR baseline and demonstrating the value of refining pre-extracted features.

## 2. Related Works

### 2.1. Latent Representation Learning

Self-supervised learning has emerged as a significant approach for representation learning, especially in scenarios with limited labeled data. This method has gained attention recently due to the increasing availability of data. Numerous self-supervised representation learning techniques have been proposed [15,16,17,18,19,20,21,25].

Traditional representation learning methods often involve applying augmentation to the original images or videos to generate similar data from various perspectives, which are then compared [15,16,17,18,19,20]. Another common approach involves masking specific parts of the input data and predicting the masked portions at the pixel or token level [21,22]. However, these methods have several limitations. The augmentation process can introduce biases and may not generalize well to other tasks or modalities [12]. Additionally, predicting masked portions can result in lower semantic-level representations [12].

Joint embedding predictive architecture (JEPA) models introduce a new approach to representation learning [11,12,14]. These models predict the representations of multiple target blocks from a single context block. This method has several advantages, as follows: it eliminates the need for separate augmentation to select positive and negative samples, pretrained encoders, human augmentation, and pixel-level reconstruction. Experimental results demonstrate that JEPA models achieve strong performance without these additional processes.

Unlike traditional methods that reconstruct masked pixel regions, JEPA models predict latent representations, making them well-suited for leveraging pre-extracted feature data. Inspired by JEPA models, this paper proposes applying a method based on JEPA’s principles to predict latent representations from pre-extracted multimodal feature data.

### 2.2. Video Moment Retrieval

To apply the method proposed in this paper, we adopted the video moment retrieval (VMR) task. VMR is a task that involves identifying a video’s specific segment corresponding to a given text query. Although various studies have been conducted on VMR, this paper utilizes models based on DETR [26]. DETR [26] is a recent end-to-end object detector based on the vision transformer, which leverages the transformer architecture for computer vision tasks. As DETR [26] has evolved, numerous downstream tasks applying the DETR framework to images, videos, and multimodal data have been extensively researched [1,2,3,4,27,28,29]. Among these, several methodologies have approached the VMR task using multimodal data, treating video moment retrieval as an object detection problem and performing both moment retrieval and highlight detection [1,2,3,4].

The datasets used by DETR-based VMR models include QVHighlight [1], which provides clip-level feature data, human-written text queries, and saliency scores for the clips. This dataset is well-suited for our approach as it offers pre-extracted clip-level feature data with text queries. We validate the efficiency of our proposed method using the QD-DETR [2] model with the QVHighlight dataset [1].

More recently, the field has seen a significant shift towards leveraging large language models (LLMs) to create powerful multimodal systems for video understanding. Models such as LLaVA-MR [23] and UniVTG [30] adapt large vision-language architectures specifically for the temporal grounding task. Others, like the large-scale foundation model InternVideo2 [24], aim for a general video comprehension capability that also proves effective for VMR. These approaches typically achieve state-of-the-art performance by capitalizing on the advanced reasoning and contextual understanding of LLMs, as we benchmark in Section 4.3. However, this performance often comes at the cost of substantial computational resources due to their massive parameter counts. Our work is positioned as a lightweight alternative, focusing on efficiently enhancing features for strong baseline models rather than employing a full-scale LLM.

## 3. Methods

This section explains the MLRL (multimodal latent representation learning) framework proposed in this paper. The overall framework is illustrated in Figure 2. Our framework takes pre-extracted clip features and text query features as input. Throughout the diagram, N represents the number of clips, L is the number of tokens in the query, and d denotes the feature dimension. The model’s primary objective is to enhance the performance of downstream tasks (in this case, video moment retrieval, VMR) by applying new representation learning techniques to multimodal data (video clip features and text query features). The overall learning process of the model is as follows: (1) The model receives pre-extracted video clip features and text queries as input data and combines the two datasets. (2) The combined data undergoes additional augmentation processing. (3) The augmented data is then divided into context blocks and target blocks. (4) Using a predictor, target blocks are predicted from the context blocks. (5) The predicted target blocks are compared with the actual target blocks to calculate the loss. (6) Based on the newly learned feature data, the VMR task is conducted. The following subsections will provide a detailed explanation of each step.

In this subsection, we explain the process of combining the input multimodal data. The proposed model takes pre-extracted multimodal feature data as input. We use the QVHighlight dataset [1], which provides feature data for each clip obtained by segmenting the original video into 2-s clips. Additionally, the dataset includes text queries corresponding to each video. Detailed information about the dataset is provided in the Dataset section.

The input data consists of clip video features and text queries. We denote the video feature data for a single video as $v=\{v_{1},\cdots ,v_{N}\}$. N represents the number of clips, each with a feature vector of 2818 dimensions. The text query data is denoted as $t=\{t_{1},\cdots ,t_{L}\}$, where L represents the number of words in a query, each with a feature vector of 512 dimensions. Before combining the two datasets, a simple fully connected network maps both data types to the same dimension. The mapped data is then concatenated to form s. This s is processed through an encoder with cross-attention (${\mathrm{ENC}}_{cross}$) to form $s_{new}$. The encoder used here is the same as that used in the VMR task after learning the latent representation. The overall process is illustrated in Equation (Equation 1):(1)$$\begin{array}{c} \begin{array}{c} v=\left\{v_{1},\cdots ,v_{N}\right\},\;t=\left\{t_{1},\cdots ,t_{L}\right\}\\ v_{map}=FFN\left(v\right),\;t_{map}=FFN\left(t\right)\\ s=\mathrm{Concat}\left(v_{map},t_{map}\right)=\left\{v_{1},\cdots ,v_{N},t_{1},\cdots ,t_{L}\right\}\\ s_{new}={\mathrm{ENC}}_{cross}\left(s\right) \end{array} \end{array}$$

The sampling process for creating context and target blocks from the combined feature sequence $s_{new}$ follows a clear, two-step procedure. First, a pool of target candidates is formed by randomly sampling a fixed ratio of α=0.15 of the total clip features without replacement, from which M individual target blocks are constructed. Next, the context block is formed from the remaining clip features that were not selected as target candidates. From this set of non-target patches, we further sample a subset at a rate of β=0.85 to form the final context block. These ratios are adopted from the I-JEPA [12] framework, where they were shown to be empirically optimal. This two-step strategy ensures that there is no overlap between the context and the information it is trying to predict, which is crucial for a meaningful learning task.

### 3.1. Target

In this subsection, we explain how to form target blocks using the combined modality data obtained in the previous step. Target blocks ($T_{i}$) refer to the parts that need to be predicted through the context block, effectively serving as label data. The fundamental idea of this predictive learning task is to force the model to learn high-level semantic representations by predicting the latent representation of a part of the input from another part. The method for constructing target blocks is as follows.

From the input $s_{new}$, a new latent representation $s_{t}$ is extracted using the target encoder $f_{t}$. This target encoder, $f_{t}$, acts as a “teacher” model, providing a stable and consistent semantic representation that serves as the prediction objective. By using a slowly evolving teacher, we prevent the model from collapsing to a trivial solution. From $s_{t}$, the part corresponding to the video length *N* is extracted to form $s_{vid}$. Target blocks are composed of a total of *M* blocks($T=\{t_{1},\cdots ,t_{M}\}$), where t refers to one target block. The sampling of target patches from $s_{vid}$ follows the two-step procedure (with a ratio of α=0.15) described at the beginning of Section 3. It is important to note that sampling for target blocks is not done directly from $s_{new}$ but from $s_{vid}$ obtained through the target encoder. This ensures the prediction task focuses on the visual features, guided by the text query information already fused into $s_{new}$. We do not apply any augmentation to the target blocks to ensure that the prediction goal remains a clean, undistorted representation of the original features. This detailed process ensures the accurate formation of target blocks, crucial for the subsequent steps in the representation learning framework.

All processes are repeated *M* times to form a total of $T_{i}$ target blocks. Target blocks may overlap with each other. The indices of the patches used in the target blocks are stored separately for later use when designating context blocks. The target encoder is a teacher encoder, identical to the context encoder used for context blocks. It is a copy of the context encoder. The target encoder is not updated by backpropagation. Instead, the parameters are updated using an exponential moving average applied to the student encoder $f_{s}$, which is used later for the context blocks.

The use of a student-teacher architecture, where the context encoder acts as the student and the target encoder as the teacher, is crucial for preventing model collapse during representation learning. If a single encoder were used to generate both context and target representations, the model could easily learn a trivial solution by simply copying the input. The teacher encoder, which is updated slowly via an exponential moving average (EMA) of the student encoder’s weights, provides a stable and consistent representation for the student to predict. This forces the student to learn higher-level semantic features rather than a simple identity function, a well-established technique to avoid collapse in self-supervised learning.

The overall process is illustrated in Equation (Equation 2).(2)$$\begin{array}{c} \begin{array}{c} s_{t}=f_{t}\left(s_{new}\right)\\ s_{vid}=s_{t}\left[:N\right]=\left\{s_{vid1},\cdots ,s_{vidN}\right\}\\ t_{i}=\mathrm{RandomSample}\left(s_{vid},\alpha N\right)\\ T=\{T_{i},\cdots ,T_{M}\} \end{array} \end{array}$$

### 3.2. Context

In this subsection, we explain how to form context blocks, which are used to predict the target blocks. The context block (*C*) provides the information given to the predictor during the prediction phase and is used to create the predicted target block for comparison with the actual target block.

The method used in previous research [12] involves extracting features directly from the original image without applying any augmentation. However, since our work uses pre-extracted features, this approach may impose some limitations on forming patches compared to methods that operate on raw data. Our proposed model applies slight augmentations to the pre-extracted data to address these limitations. The augmentations include adding Gaussian noise and applying feature dropout. These augmentations are applied only when forming the context block, not the target block. This one-sided augmentation strategy is a core component of our approach. By corrupting the context, we create a more challenging prediction task, forcing the model to learn robust and generalizable features rather than simply memorizing the input data.

The method for forming context blocks is similar to the method for forming target blocks. From the input $s_{new}$, only the part corresponding to the video is used (denoted as $c_{vid}$). The context block is formed by taking the remaining parts of $c_{vid}$ after excluding the patch indices used for the target blocks. From these remaining patches, the final context block is sampled with a rate of β=0.85, as detailed in the procedure at the start of Section 3. Finally, the previously described augmentations are applied to the constructed context patches, and then the context encoder ($f_{c}$) is applied to form the context block (*C*). Unlike the target blocks, only one context block is formed, and this context block is used to predict *M* target blocks. The context encoder used here is the student encoder, which is updated through backpropagation. The context encoder is also used later in the VMR task as an encoder. The entire process is illustrated in Equation (Equation 3).(3)$$\begin{array}{c} \begin{array}{c} c_{vid}=s_{new}\left[:N\right]=\left\{v_{1},\cdots ,v_{N}\right\}\\ C=\mathrm{RandomSample}(c_{vid},\beta N)\\ C=f_{c}\left(Aug\left(C\right)\right) \end{array} \end{array}$$

### 3.3. Latent Representation Prediction

This section describes the method for extracting *M* predicted target block representations using the context block (*C*) formed in the previous stage. Each predicted target block ($\hat{T}$) representation corresponds to a target block (*T*) that was established earlier. The method for predicting the target blocks is as follows. The input is *C* obtained from the context encoder and mask tokens (MASK). The number of mask tokens corresponds to the number of target blocks that need to be predicted. ($\mathrm{MASK}=\{{\mathrm{mask}}_{1},\cdots ,{\mathrm{mask}}_{J}\}$) J denotes the number of target patches in one target block. Each mask token is a learnable parameter and has an added positional embedding. Context *C* and the mask tokens [MASK] are fed into the predictor ($f_{p}$) to obtain the predicted representation (${\hat{T}}_{i}$) for each target block ($T_{i}$). The predictor’s role is to synthesize the latent representation of the target by attending to the most relevant information within the context block C. Since predictions need to be made for *M* target blocks, this process is repeated *M* times.

In this stage, the predictor adopts the same DETR-style object-query decoder used in the downstream VMR head; we simply warm-start the VMR decoder with the weights learned during latent prediction [31]. While the encoders used in the previous stage are the same as those employed later in the VMR task, the decoder used here differs from the one used in the VMR task. The decoder is trained through backpropagation while representation learning. The overall process of latent representation prediction is illustrated in Equation (Equation 4):(4)$$\begin{array}{c} \begin{array}{c} \mathrm{MASK}=\{{\mathrm{mask}}_{i},\cdots ,{\mathrm{mask}}_{J}\}\\ {\hat{T}}_{i}=f_{p}\left(C,\mathrm{MASK}\right)\\ \hat{T}=\left\{{\hat{T}}_{i},\cdots ,{\hat{T}}_{M}\right\} \end{array} \end{array}$$

### 3.4. Loss Function

This section explains the loss function used in latent representation learning. The loss function, as shown in Equation (Equation 5), calculates the mean squared error (MSE) between the latent representation of the target block (*T*) and the predicted latent representation of the target block ($\hat{T}$). We use MSE as it provides a direct and efficient way to minimize the Euclidean distance between the predicted and target vectors in the latent space. This encourages the predictor to generate representations that are not only semantically similar but also close in their geometric arrangement.(5)$$\begin{array}{c} MSE=\frac{1}{M}\sum_{i=1}^{M}\left({\hat{T}}_{i}-T_{i}\right) \end{array}$$

The calculated loss is used to update the predictor and context encoder parameters through backpropagation. The parameters of the target encoder are not updated through backpropagation of the loss; instead, they are updated using the exponential moving average of the context encoder’s parameters [15,32].

### 3.5. Video Moment Retrieval

After completing the first stage of latent representation learning, the model proceeds to the main task, that is, video moment retrieval (VMR). VMR involves identifying the segment of a video that corresponds to a given text query. The general framework for this stage, based on the QD-DETR [2] model, is illustrated in Figure 3.

The input consists of video clip features and text queries. Using data from both modalities, the model ultimately predicts the relevant moment in the video as an object detection output, specifying the middle timestamp and the span on either side. Additionally, QD-DETR [2] performs highlight detection using the saliency scores provided by the QVHighlight dataset [1].

The method proposed in this paper aims to verify the performance changes in downstream tasks when additional representation learning is applied to pre-extracted feature data. Therefore, the training method for the VMR stage remains unchanged. As shown in Figure 3, the key difference in our approach is that the original T2V encoder and main encoder of the QD-DETR [2] model are replaced with the cross-encoder and context encoder learned during the MLRL stage.

## 4. Results

### 4.1. Dataset and Metrics

This paper uses the QVHighlight [1] dataset, which provides labeled data for video moment retrieval and highlight detection tasks. The dataset consists of 10,148 videos sourced from YouTube, accompanied by 10,310 human-written text queries. Following the official splits, the dataset is divided into 7088 videos for the training set, 1518 for the validation set, and 1542 for the test set.

Each video is approximately 150 s long, and the queries average 11.3 words. For our experiments, we use the pre-extracted features provided by the original authors. Specifically, the video features are 2304-dimensional vectors extracted from a pre-trained SlowFast network, and the query features are 768-dimensional vectors from a pre-trained RoBERTa model. The original video data is divided into 2 s clips for this feature extraction. Furthermore, the dataset provides saliency scores for each clip, indicating its relevance to the query, which is used for the highlight detection task.

To validate the generalizability of our approach, we also evaluate its performance on the Charades-STA dataset [6], a widely-used benchmark for sentence-based temporal activity localization. Derived from the original Charades dataset, it consists of 12,408 temporal annotations across 6672 training videos and 3720 annotations for 1334 testing videos. For fair performance analysis, we used the same metrics as the baseline QD-DETR [2] to measure performance. We measured performance using recall and mean average precision (mAP). Recall was measured at IoU thresholds of 0.5 and 0.7; mAP was measured at IoU values of 0.5, 0.7, and as the average over multiple thresholds in the range [0.5:0.95; interval = 0.05].

### 4.2. Experiment Setup

The proposed method in this paper, MLRL, is divided into two stages. One is the latent representation learning stage, where additional representation learning is conducted on the pre-extracted feature data proposed in this paper. The other is the video moment retrieval stage, where the downstream task is performed. This paper proposes a methodology for the latent representation learning stage and, consequently, does not significantly modify the moment retrieval stage. Therefore, most settings are the same as the baseline QD-DETR [2]. In this paper, experiments were conducted on two scenarios. One scenario involves conducting representation learning and moment retrieval separately, and the other consists of conducting both simultaneously. In the simultaneous scenario, the settings for representation learning are the same as QD-DETR [2]. In contrast, in the separate scenario, the settings are almost identical to those in I-JEPA [12]. The differences are that the batch size is set to 32 and the epochs are set to 100. The separate regime reflects a practical use case in which the encoder is refined once on a large, unlabeled corpus and then reused across multiple retrieval heads without rerunning the expensive detector, thereby reducing GPU-hours and memory peaks. The main settings in the latent representation learning stage are as follows. The hyperparameters for sampling, namely the target block ratio α and the context block ratio β, were set to 0.15 and 0.85, respectively. This decision is based on the optimal values presented in the original I-JEPA [12] paper, where extensive ablation studies (Assran et al., 2023 [12]) confirmed that this configuration provides the best performance by balancing the complexity of the prediction task and the informativeness of the context. The weights of the target encoder are the same as those of the context encoder. At this time, the target encoder is updated not through backpropagation but through an exponential moving average (EMA). The momentum of the EMA increases linearly from 0.996 to 1.0 during the training process. All experiments were conducted in an environment equipped with two Intel(R) Xeon(R) Gold 6248R CPUs, 345GiB of RAM, and eight NVIDIA TITAN RTX GPUs (each with 24 GB VRAM).

### 4.3. Experiment Result

This section introduces the experimental results for MLRL. The experiments compare the performance of MLRL with the baseline [2] and the original version of [1,2]. Table 1 shows the experimental results.

Before explaining the experimental results, let us first define the terms used in the table. The mAP in the table is the average value of IoU thresholds from 0.5 to 0.95. R represents the recall value. Extra training data indicates whether other datasets were used in addition to [1] during the training process. Rep/MR means that latent representation learning and moment retrieval were conducted separately. This means the encoders were trained through representation learning first and then used in moment retrieval. Rep+MR means that latent representation learning and moment retrieval were conducted simultaneously. The final loss is calculated by combining the loss of the representation learning method and the loss of the moment retrieval method within one framework.

Comparing the results with the baseline QD-DETR(val) in Table 1, our method shows clear performance gains. The Rep/MR model, for instance, achieved an increase of +1.05 in mAP and +0.89 in R1@0.7. This demonstrates that enhancing feature representation through our proposed MLRL is more effective than simply using pre-extracted features for the downstream task.

To provide a broader context for our work and address the need for additional benchmarks, we also compare our method with recent state-of-the-art models that leverage large language models (LLMs). The results are presented in Table 2. As shown in Table 2, recent state-of-the-art models leveraging large language models (LLMs) demonstrate high performance on the QVHighlight benchmark. However, a direct comparison of raw performance metrics with our MLRL framework requires careful consideration of the models’ scale and design philosophy. These LLM-based models [23,24,30,33], such as LLaVA-MR [23] and InternVideo2-6B [24], often contain billions of trainable parameters and demand substantial computational resources for training and inference. In contrast, the primary goal of our MLRL is not to compete with these massive models in raw performance, but to provide a lightweight and efficient method for significantly enhancing pre-extracted features with minimal overhead. Therefore, the results in Table 2 should be interpreted from two perspectives, as follows: while LLMs define the upper bound of performance on this benchmark, our MLRL framework demonstrates a more practical and resource-efficient pathway to achieving substantial improvements over strong, non-LLM baselines like QD-DETR [2]. This highlights the value of our approach for scenarios where computational efficiency is a critical consideration.

To validate its generalizability, we also evaluated our framework on the Charades-STA [6] dataset, with results in Table 3. On this dataset, our MLRL(Rep+MR) model improves upon the QD-DETR baseline by +0.70 points on R1@0.5 and a more significant +1.99 points on R1@0.7. These results confirm that the benefits of MLRL extend to other standard benchmarks, proving its robustness.

### 4.4. Ablation Study

In this section, we compare the results based on different modalities of data used for representation learning through the proposed MLRL. The comparison is conducted across four modalities, as follows: Clip/Query, Clip, Query, and None. ‘Clip/Query’ refers to the case where both clip features and query features are utilized together, ‘Clip’ refers to the case where only clip features are used, ‘Query’ refers to the case where only query features are used, and ’None’ represents the baseline where latent representation learning is not applied. The cross-encoder, which combines the two modalities, is not applied to the Clip and Query cases. In the case of Query, since there is no clip, a mask is added to the tokens during context and target sampling.

As shown in Table 4, the results show that using Clip/Query or Clip alone leads to an improvement in metrics. Specifically, for Clip/Query, there was an increase of +1.05 in mAP, +0.72 in mAP@0.75, and +0.89 in R1@0.7. For Clip, the increases were +0.95 in mAP, +0.23 in mAP@0.5, +0.26 in mAP@0.75, and +0.65 in R1@0.7. On the other hand, when using only Query, the metrics tend to be similar to or even lower than when no features were used at all.

In summary, the results indicate that using Clip/Query or Clip leads to a noticeable improvement in performance, highlighting the critical role of visual information (clip features) in video moment retrieval tasks. When both clip and query features are used together, performance improves at higher thresholds, suggesting that combining visual and textual information can have a complementary effect in enhancing model accuracy. On the other hand, when only query features are used, performance does not improve significantly or may even decline, indicating that textual information alone may not be sufficient for this task. Overall, these findings emphasize the importance of integrating visual information and multimodal data to achieve more robust and accurate results in video moment retrieval.

### 4.5. Qualitative Result

In this section, we examine how the performance of the proposed MLRL model is reflected in actual qualitative results. Figure 4 shows the ground truth (G.T.), the predicted segments from MLRL, and the predicted segments from QD-DETR [2] for a portion of the test. Upon review, it is evident that both QD-DETR [2] and MLRL predict segments similarly to the G.T. The key difference is that while QD-DETR [2] separates the segments like the G.T., MLRL predicts a single, combined segment. However, upon closer inspection of the G.T., we find that the portion where the segments are split, which corresponds to the part not related to the query (a close-up of a woman’s face), lasts for only about one second and is semantically part of a single segment. In this sense, it can be concluded that MLRL, through additional training that enhances the video’s representation, is capable of segment prediction that considers the semantic aspects of the video.

## 5. Discussion

In this section, we discuss the issues discovered during the experiments with the model and potential future research directions. One problem identified during the experiments is the ambiguous utilization of query features in the representation learning stage. In the current method, query features are used by concatenating clip features and query features and then inputting them into a cross-encoder to transfer the information of query features to clip features. We believe that using a method where the query directly influences the loss calculation in representation learning can result in better performance in the multimodal task of moment retrieval.

Based on the current research, we introduce potential future work. Research could explore ways to make query features have a more direct impact on representation learning, rather than just using them in cross-attention. For instance, a method could be proposed in which context blocks and target blocks are selected based on their relevance to the query.

Another direction for future research could be to demonstrate the agnostic nature of the model. The method proposed in this study pertains to representation learning and is an agnostic approach that can be applied not only to the video moment retrieval (VMR) method used in this study, but also to other VMR methods [3,4,5]. Therefore, future research could involve applying the proposed method to other VMR methods to verify that the method is indeed model agnostic.

## 6. Conclusions

This paper addresses the challenge of processing large-scale video sensor data, where direct feature extraction from raw signals is a major computational bottleneck. We propose the multimodal latent representation learning (MLRL) framework, which enhances the expressiveness of pre-extracted features through an additional representation learning stage. Experiments confirmed that our model significantly outperforms approaches that simply use the pre-extracted features for downstream tasks. This result indicates that MLRL offers a practical pathway to more accurately interpret sensor data without costly reprocessing, thereby contributing to the development of more efficient and powerful sensor systems. While the effectiveness of MLRL has been demonstrated, there remains room for future research, such as improving the utilization of query features and proving the model’s agnostic nature across different architectures.

## Figures and Tables

**Figure 1 sensors-25-04528-f001:**
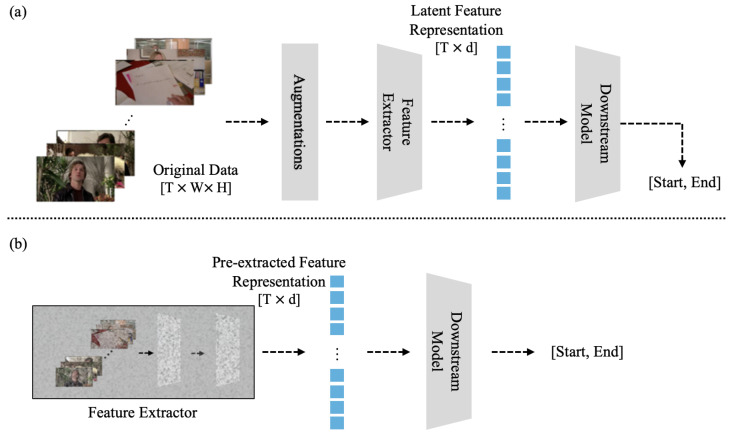
**Limitations of the video learning process.** Comparison of video processing pipelines. (**a**) The standard training process includes feature extraction from the original data. (**b**) Our simplified process leverages pre-extracted features.

**Figure 2 sensors-25-04528-f002:**
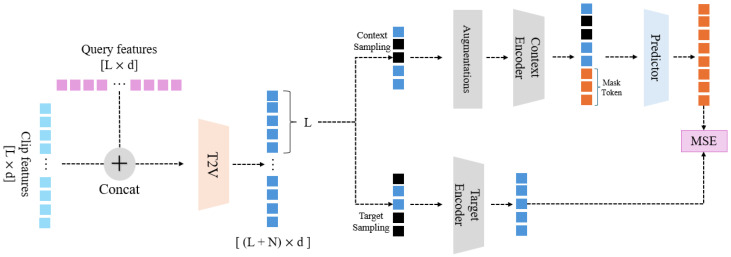
**Overview of the latent representation learning process.** An overview of our proposed multimodal latent representation learning (MLRL) framework.

**Figure 3 sensors-25-04528-f003:**
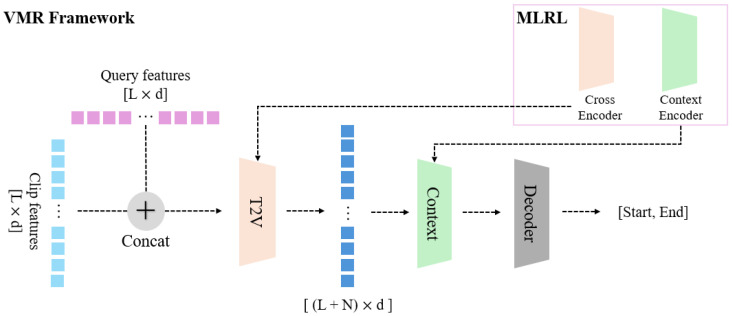
**Video moment retrieval framework.** The video moment retrieval (VMR) framework, based on QD-DETR [2]. Our proposed MLRL module (top right) replaces the original encoders.

**Figure 4 sensors-25-04528-f004:**
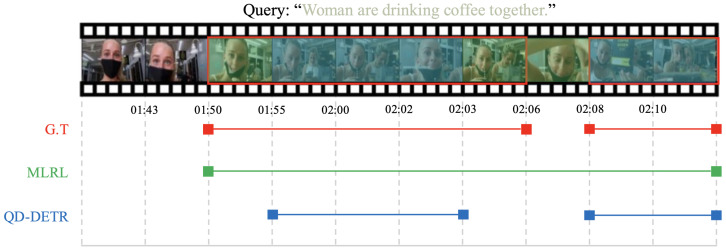
**Qualitative result.** This image shows the results of a qualitative experiment comparing MLRL with ground truth (G.T.) and QD-DETR.

**Table 1 sensors-25-04528-t001:** **Comparison results of the multimodal latent representation learning (MLRL).** This table compares the performance of MLRL with previous models, Moment-DETR [1] and QD-DETR [2]. The term (test) refers to evaluations conducted on the test dataset of QVHighlight [1], while (val) indicates evaluations performed on the validation dataset. The performance of MLRL was assessed using the validation set of the QVHighlight [1] dataset.

Model	mAP	mAP@0.5	mAP@0.75	R1@0.5	R1@0.7	Extra Data
Moment-DETR [1]	30.73	54.82	29.40	52.89	33.02	x
QD-DETR(w/audio) [2]	40.19	63.04	40.10	63.06	45.10	x
QD-DETR(test) [2]	39.86	62.52	40.10	62.40	45.10	x
QD-DETR(val) [2]	41.22	62.23	41.82	62.68	46.66	x
MLRL(Rep/MR)	**42.27**	62.20	**42.54**	62.26	47.55	x
MLRL(Rep+MR)	42.06	**62.32**	42.06	**62.32**	**47.65**	x

**Table 2 sensors-25-04528-t002:** **Performance comparison with LLM-based models on the QVHighlight dataset.** This table compares our MLRL models against recent LLM-based methods [23,24,30,33]. All models are evaluated on the QVHighlight validation set without external training data. The best performance for each metric is underlined.

Model	mAP	mAP@0.5	mAP@0.75	R1@0.5	R1@0.7
UniVTG (w/PT)	43.63	64.06	45.02	65.43	50.06
InternVideo2-6B	49.24	-	-	71.42	56.45
FlashVTG	52.00	72.33	53.85	70.69	53.96
LLaVA-MR	52.73	69.41	54.40	76.59	61.48
MLRL(Rep/MR)	**42.27**	62.20	**42.54**	62.26	47.55
MLRL(Rep+MR)	42.06	**62.32**	42.06	**62.32**	**47.65**

**Table 3 sensors-25-04528-t003:** **Performance comparison on the Charades-STA dataset.** Performance on the Charades-STA [6] dataset to evaluate generalizability. ‘w/PT’ indicates the Moment-DETR [1] model was pre-trained on 10K HowTo100M videos.

Model	R1@0.5	R1@0.7
Moment-DETR [1]	53.63	31.37
Moment-DETR [1] w/PT (on 10K HowTo100M videos)	55.65	34.17
QD-DETR [2] (Only Video)	57.31	32.55
MLRL(Rep/MR)	57.96	**33.77**
MLRL(Rep+MR)	**58.01**	33.65

**Table 4 sensors-25-04528-t004:** **Ablation study of the latent representation learning.** This table presents the results when applying latent representation learning to different modalities. ‘Clip/Query’ shows the results when both clip features and query features are used together, ‘Clip’ shows the result when only clip features are used, and ‘None’ represents the results without applying latent representation learning.

Modality	mAP	mAP@0.5	mAP@0.75	R1@0.5	R1@0.7
Clip/Query	42.27	62.20	42.54	62.26	47.55
Clip	42.17	62.46	42.08	62.35	47.31
Query	41.03	61.97	41.65	62.61	46.23
None(Baseline)	41.22	62.23	41.82	62.68	46.66

## Data Availability

The data presented in this study are openly available in https://github.com/jayleicn/moment_detr, accessed on 15 January 2025.

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
