# Peer review of "Multimodal Latent Representation Learning for Video Moment Retrieval"

_sensors, 2025, doi:10.3390/s25144528_

Round 1

Reviewer 1 Report

Comments and Suggestions for Authors
  • The introduction should've been better with slightly more structured explanation distinguishing clearly between general trends (e.g., LLMs) and the specific technical gap this work addresses.
  • Most parts are well explained, particularly with Figures 2 and 3 that visually support understanding.
  • However, some equations and process descriptions (e.g., how context and target blocks are sampled) could be more concise and clarified. For instance:
    1. The hyperparameters α and β are introduced without first framing how they are chosen.
    2. The logic behind using different encoders (student vs teacher) is briefly mentioned but could be more clearly justified.
  • A minor suggestion: standardize figure and table captions — some are more descriptive than others.
  • Suggestion for future work: improve query feature usage and demonstrate model agnosticism, as these efforts are appropriate and open potential avenues for continued research.
  • Figure 4 (qualitative results) lacks detail and might benefit from clearer visual markers (e.g., timestamps, colored highlights).
Comments on the Quality of English Language

The manuscript is understandable but could be polished for grammar and clarity. Some repeated phrases, long sentences, and awkward constructions should be revised.

Author Response

Comment 1: The introduction should've been better with a slightly more structured explanation, distinguishing clearly between general trends (e.g., LLMs) and the specific technical gap this work addresses.

Response 1: Thank you for the constructive feedback. We agree that a more structured introduction would improve the clarity of our paper. Accordingly, we have completely rewritten the Introduction (Section 1) to establish a clear logical flow. The revised introduction now begins by presenting the general problem, then narrows down to the specific technical gap this paper addresses, and finally presents our MLRL framework as a direct solution to that gap. This new structure better distinguishes our specific contribution from general industry trends. The changes can be found throughout Section 1.

Comment 2: Most parts are well explained, particularly with Figures 2 and 3 that visually support understanding. However, some equations and process descriptions (e.g., how context and target blocks are sampled) could be more concise and clarified.

Response 2: Thank you for pointing this out. We agree that the original description of the sampling process was scattered across multiple sections and could be made more concise and clear. To address this, we have made the following revisions:

  1. First, we have added a new, unified paragraph at the beginning of the Methods (Section 3) that clearly explains our two-step sampling procedure for context and target blocks. This consolidated description allows readers to understand the entire process in one place. This change can be found in Section 3.
  2. Consequently, we have removed the redundant descriptions of the sampling ratios from the Target (now Section 3.2) and Context (now Section 3.3) subsections to improve conciseness.
  3. Finally, we have replaced the deleted sentences with brief "pointer" sentences that refer back to the main sampling description, ensuring a clear logical flow. These changes are outlined in Sections 3.2 and 3.3.

We believe these changes make our methodology much easier to follow and fully address the reviewer's concerns regarding clarity and conciseness.

Comment 3 (related to Comment 2): The hyperparameters α and β are introduced without first framing how they are chosen.

Response 3: We agree that the rationale for choosing the hyperparameters should be clarified. Our choice of the target block sampling rate (alpha=0.15) and context block sampling rate (beta=0.85) directly follows the experimentally validated, optimal settings from the foundational I-JEPA framework (Assran et al., 2023), which our work is built upon.

Experiment Setup.

”The hyperparameters for sampling, namely the target block ratio (alpha) and the context block ratio (beta), were set to 0.15 and 0.85, respectively. This decision is based on the optimal values presented in the original I-JEPA paper, where extensive ablation studies (Assran et al., 2023) confirmed that this configuration provides the best performance by balancing the complexity of the prediction task and the informativeness of the context.”

Comment 4:The logic behind using different encoders (student vs teacher) is briefly mentioned, but could be more clearly justified.

Response 4: Thank you for this important suggestion. To better justify our use of the student-teacher architecture, we have added a detailed explanation to the Methods section (Section 3.1), immediately following the introduction of the teacher encoder. The new text clarifies that this architecture is a crucial technique to prevent model collapse in self-supervised learning. The added text is as follows:

The use of a student-teacher architecture, where the context encoder acts as the student and the target encoder as the teacher, is crucial for preventing model collapse during representation learning. If a single encoder were used to generate both context and target representations, the model could easily learn a trivial solution by simply copying the input. The teacher encoder, which is updated slowly via an Exponential Moving Average (EMA) of the student encoder's weights, provides a stable and consistent representation for the student to predict. This forces the student to learn higher-level semantic features rather than a simple identity function, a well-established technique to avoid collapse in self-supervised learning.”

Comment 5: A minor suggestion: standardize figure and table captions — some are more descriptive than others.

Response 5: Thank you for this helpful suggestion. We have reviewed all the figure and table captions throughout the manuscript and revised them for clarity and consistency. Detailed descriptions have been moved from the captions to the main body text to improve readability, particularly for Figure 1 and Figure 2.

Comment 6: Suggestion for future work: improve query feature usage and demonstrate model agnosticism, as these efforts are appropriate and open potential avenues for continued research.

Response 6: We thank the reviewer for their positive feedback and for validating our proposed directions for future work. We are encouraged by their assessment.

Comment 7: Figure 4 (qualitative results) lacks detail and might benefit from clearer visual markers (e.g., timestamps, colored highlights).

Response 7: We agree that the qualitative results in Figure 4 could be presented more clearly. We have revised Figure 4 to include clearer timestamps and have used distinct colored highlights overlaid on the timeline to better distinguish the predictions of our model, the baseline, and the ground truth. We believe the revised figure is now much more detailed and easier to interpret.

Reviewer 2 Report

Comments and Suggestions for Authors

Title: Multi-modal Latent Representation Learning for Video Moment Retrieval

In this work, a novel method called Multi-modal Latent Representation Learning framework (MLRL) is proposed that enhances the performance of downstream tasks by conducting additional representation learning on pre-extracted features. The work is good; however, this manuscript currently has major and minor corrections (described below), that should be considered carefully.

Remarks to the Author: Please see the full comments.

1- A benchmark comparison of the proposed model with the QD-DETR model was performed. However, it is recommended to compare the proposed work against other existing models to confirm its effectiveness. This is an important point in every research paper to show the noticeable improvement compared to other work in the same field and to examine the performance.

In addition, the word “novel” can be replaced with another synonym.

2- How has the potential of MLRL to simplify video data processing been evaluated, and what is the rate of improvement in the accuracy of the proposed model?

3- It is recommended to shorten the captions of the figures and transfer their explanation to the body of the text, as that in Figures 1 and 2.

4- It is stated in lines (51-53) that “In this paper, we praose a new method called Multimodal Latent Representation Learning (MLRL). This method enhances the performance of downstream tasks by conducting additional representation learning on pre-extracted features.”.  First, the word “praose” should be corrected. Second, this point should be added to the contributions.

Finally, the contributions mentioned are weak and should be reviewed to highlight the main points of research and the new idea that is added to this field.

5- The manuscript contains some grammatical and linguistic errors, and these errors lead to a decrease in the reader's understanding, so the entire manuscript should be reviewed carefully. In addition, there are some points that should be corrected which are:

- In Figure 1, the word “Featrue” should be corrected to be “Feature”.

-In line “344” the number of the figure is missed.

-For more organization, the point should be placed after the reference number. For example, please check line “222”.

-The title of the figure should be above the Table such as in Table 1.

-What is the purpose of the last column “Extra Data” in Table 1.

6- The introduction and related work sections provide a comprehensive foundation on the topic of this research based on various works; however, some other background and recent works (in the years 2024 and 2025) need to be added to this section with much in-depth discussion.

7- Any information, graph, equation, or dataset taken from a previous source must be documented with a reliable source, unless it belongs to the authors. Please check this issue for the full manuscript.

8- The adoption of the Video Moment Retrieval (VMR) task needs further justification since it is related to the proposed system model.

9- While the time-intensive feature extraction process required for model training is mentioned in the Abstract is an important limitation, the hardware specifications of the system used to measure this issue are not provided in the manuscript. Including these details is essential for reproducibility and benchmarking.

10- Can the authors please provide further clarification and details about the datasets used in this paper.  

11- Why are the parameters of the target encoder updated using the exponential moving average (EMA). Please explain.

12- It is advisable to rewrite the conclusion section into one coherent paragraph.

Comments on the Quality of English Language

The English should be improved to more clearly express the research

Author Response

Comments1: A benchmark comparison of the proposed model with the QD-DETR model was performed. However, it is recommended to compare the proposed work against other existing models to confirm its effectiveness. This is an important point in every research paper to show the noticeable improvement compared to other work in the same field and to examine the performance. In addition, the word “novel” can be replaced with another synonym.

Response 1: Thank you for your insightful comments. We agree that providing a broader comparison with other state-of-the-art models is crucial for contextualizing our work, and we appreciate the suggestion regarding word choice. Therefore, we have revised the manuscript to include additional benchmarks and have refined our terminology.

  1. Addition of SOTA Benchmarks and Dataset: To address your primary comment, we have expanded our experimental validation in two significant ways:
  • First, we added a new benchmark comparison against recent, powerful LLM-based models on the QVHighlight dataset to position our work within the current state-of-the-art landscape.
  • Second, we conducted new experiments on the Charades-STA dataset to demonstrate the generalizability and effectiveness of our method beyond a single dataset.
  • Location: The new tables and corresponding analyses can be found in Section 4.3. Experiment Result.
  1. Revision of the word "novel": We have carefully reviewed the manuscript and replaced the word "novel" with more appropriate synonyms to improve the phrasing.
  • Location: Abstract (Page 1)
  • Original Text:

”We propose a novel method called Multi-modal Latent Representation Learning framework(MLRL) to address these limitations.”

  • Revised Text:

”We propose a new method called Multi-modal Latent Representation Learning framework(MLRL) to address these limitations.”

We believe these revisions substantially strengthen the paper by providing a more robust and comprehensive evaluation of our proposed method.

comments2: How has the potential of MLRL to simplify video data processing been evaluated, and what is the rate of improvement in the accuracy of the proposed model?

Response 2: Thank you for your question. We have clarified both of these points in the revised manuscript.

  1. Evaluation of Simplification Potential: The potential of MLRL to simplify video data processing was evaluated primarily by the inherent design of our framework. Our approach fundamentally simplifies the traditional pipeline by operating on pre-extracted features. This bypasses the most time-consuming and computationally expensive step: feature extraction from raw video data.
  • Location: This rationale is explained in the Introduction (Section 1) and visually contrasted in
  1. Accuracy Improvement Rate: Our model demonstrates significant accuracy improvements over the strong baseline (QD-DETR) on two different benchmark datasets.
  • Location: The detailed results are presented in Section 4.3 and summarized in Tables 1 and 3.
  • On the QVHighlight dataset, our Rep/MR model achieved an increase of +1.05 in mAP and +0.89 in R1@0.7 compared to the QD-DETR baseline.
  • On the Charades-STA dataset, to prove generalizability, our Rep+MR model improved upon the QD-DETR baseline by a significant +1.99 on R1@0.7

These results confirm that our resource-efficient approach not only simplifies the process but also effectively enhances model performance.

comments3: It is recommended to shorten the captions of the figures and transfer their explanation to the body of the text, as in Figures 1 and 2.

Response 3: Thank you for the excellent suggestion on improving the presentation of our figures. We agree that moving detailed descriptions from captions into the main text enhances readability and flow. Therefore, we have reviewed all figures in the manuscript and have revised the captions for Figures 1, 2, and 3 to be more concise, integrating their detailed explanations into the body of the text where they are first introduced.

comments4: It is stated in lines (51-53) that “In this paper, we propose a new method called Multimodal Latent Representation Learning (MLRL). This method enhances the performance of downstream tasks by conducting additional representation learning on pre-extracted features. First, the word “praose” should be corrected. Second, this point should be added to the contributions.

Response 4: Thank you for your careful review and helpful suggestions. We agree that this is a core point of our work and should be explicitly listed as a contribution. Therefore, we have corrected the typo and revised the list of contributions at the end of the Introduction to better highlight this point.

  1. Correction of Typo: The typo "praose" has been corrected to "propose".
  • Location: Introduction (Page 2)
  1. Revision of Contributions: We have rewritten the contributions list to be more specific and to directly state that our framework is designed to enhance pre-extracted features.
  • Location: End of Introduction (Page 3)
  • Original Contributions list:

”Our contributions are as follows. 1) Highlighting the limitations of using original video data. 2) Proposing a new methodology for feature representation enhancement. 3) Validating the effectiveness of this approach through empirical evaluation on video moment retrieval tasks.”

  • Revised Contributions list:

”Our contributions are as follows. 1) We propose MLRL, a framework specifically designed to enhance pre-extracted multi-modal features, addressing the performance gap left by methods that use these features without refinement. 2) We demonstrate the successful application of a JEPA-inspired latent prediction objective to a multi-modal (video-text) context, extending its typical single-modality application. 3) Our framework achieves significant performance gains on the QVHighlight benchmark, improving upon a strong QD-DETR baseline and demonstrating the value of refining pre-extracted features.”

Comments: The manuscript contains some grammatical and linguistic errors, and these errors lead to a decrease in the reader's understanding, so the entire manuscript should be reviewed carefully. In addition, some points should be corrected, which are:

  • 5-1. In Figure 1, the word “Featrue” should be corrected to be “Feature”.
  • 5-2. In line “344” the number of the figure is missed.
  • 5-3. For more organization, the point should be placed after the reference number. For example, please check line “222”.
  • 5-4. The title of the figure should be above the Table such as in Table 1.
  • 5-5. What is the purpose of the last column “Extra Data” in Table 1.

Response 5: Thank you for your thorough review and constructive feedback. We acknowledge that grammatical and linguistic errors can hinder readability. Accordingly, we have carefully revised the manuscript based on all your comments. The entire manuscript has been professionally proofread to correct grammatical errors and improve clarity.

Below are our point-by-point responses to your specific comments.

5-1. Agree. We have corrected the typo "Featrue" to "Feature".

  • Location: Figure 1 (Page 2)

5-2. Agree. We have reviewed the manuscript and ensured that all figures are correctly numbered.

5-3. Thank you for the suggestion. We have revised the manuscript to ensure that all punctuation, such as periods, is placed after the reference numbers for consistency.

5-4. Thank you for the comment. We have double-checked the manuscript and confirmed that all Table titles are located above the tables, and all Figure captions are located below the figures, which aligns with standard academic formatting.

5-5. The "Extra Data" column is included to ensure a fair and direct comparison between our proposed model and the baseline models. It indicates whether any training data other than the primary QVHighlight[1] dataset was used. As shown in Table 1, all listed models, including ours, were trained without any extra data ("X"). This demonstrates that the performance improvements are attributable to our methodology itself, not to the use of additional data sources. The explanation for this is provided in the manuscript.

  • Location: Section 4.3 Experiment Result (Page 9, Paragraph 3)
  • Text from the manuscript:

”Extra training data indicates whether other datasets were used in addition to [1] during the training process.”

comments6: The introduction and related work sections provide a comprehensive foundation on the topic of this research based on various works; however, some other background and recent works (in the years 2024 and 2025) need to be added to this section with much in-depth discussion.

Response 6: Thank you for your valuable suggestion. We agree that incorporating a discussion on the latest state-of-the-art research provides important context for our work. Accordingly, we have revised both the Introduction and the Related Works (Section 2.2) sections to include a more in-depth discussion of recent (2024) works, particularly the trend of using Large Language Models (LLMs) in this domain.

  1. Update to Introduction: We have briefly updated the Introduction to mention specific examples of recent state-of-the-art models.
  2. In-depth Discussion in Related Works: To provide a more detailed discussion as requested, we have added a new paragraph to the end of the "Video moment retrieval" subsection that discusses the recent shift towards LLM-based methods.

comments7: Any information, graph, equation, or dataset taken from a previous source must be documented with a reliable source, unless it belongs to the authors. Please check this issue for the full manuscript.

Response 7: Thank you for emphasizing this crucial point. We agree completely that all external information must be meticulously documented with reliable sources. Accordingly, we have conducted a thorough review of the entire manuscript to ensure that all information, datasets, and baseline models originating from previous works are properly cited.

comments8: The adoption of the Video Moment Retrieval (VMR) task needs further justification since it is related to the proposed system model.

Response 8: Thank you for your valuable suggestion. We agree that a more explicit justification for selecting the Video Moment Retrieval (VMR) task improves the manuscript. Accordingly, we have expanded upon our rationale in the Introduction section to better justify the choice of VMR as the primary task for evaluation. We have clarified that VMR was chosen because its core characteristics make it an ideal testbed for our MLRL framework. This justification is now more clearly stated in the manuscript.

  • Location: Introduction (Page 2, Paragraph 4)
  • Added Text:

”We specifically select the Video Moment Retrieval (VMR) task as an ideal testbed because its core characteristics align perfectly with the objectives of our MLRL framework. As an inherently multi-modal task, VMR requires a deep understanding of both visual and textual information, allowing us to directly validate our multi-modal latent prediction objective. Furthermore, the reliance of many state-of-the-art VMR models on pre-extracted features and the recent emergence of powerful LLM-based approaches in this domain make it the perfect scenario to demonstrate the value of our enhancement framework.

Comments9: While the time-intensive feature extraction process required for model training is mentioned in the Abstract is an important limitation, the hardware specifications of the system used to measure this issue are not provided in the manuscript. Including these details is essential for reproducibility and benchmarking.

Response 9: Thank you for your valuable comment. We agree that providing hardware specifications is essential for reproducibility. Accordingly, we have now added the detailed hardware environment of our experiments to the manuscript.

  • Location: Section 4.2. Experiment Setup
  • Added Text:

”All experiments were conducted in an environment equipped with two Intel(R) Xeon(R) Gold 6248R CPUs, 345GiB of RAM, and eight NVIDIA TITAN RTX GPUs (each with 24GB VRAM).”

comments10: Can the authors please provide further clarification and details about the datasets used in this paper.

Response 10: Thank you for your valuable feedback. We agree that providing more detailed information about the datasets is essential for clarity and reproducibility. Accordingly, we have significantly expanded Section 4.1. Dataset and Metrics to include more comprehensive details for both the QVHighlight and the newly added Charades-STA datasets

Comments 11: Why are the parameters of the target encoder updated using the exponential moving average (EMA). Please explain.

Response 11: To better justify our use of the student-teacher architecture, we have added a detailed explanation to the Methods section (Section 3.1), immediately after the teacher encoder is introduced. The new text clarifies that this architecture is a crucial technique to prevent model collapse in self-supervised learning. The added text is as follows:

The use of a student-teacher architecture, where the context encoder acts as the student and the target encoder as the teacher, is crucial for preventing model collapse during representation learning. If a single encoder were used to generate both context and target representations, the model could easily learn a trivial solution by simply copying the input. The teacher encoder, which is updated slowly via an Exponential Moving Average (EMA) of the student encoder's weights, provides a stable and consistent representation for the student to predict. This forces the student to learn higher-level semantic features rather than a simple identity function, a well-established technique to avoid collapse in self-supervised learning.”

Comments 12: It is advisable to rewrite the conclusion section into one coherent paragraph.

Response 12: Thank you for the excellent suggestion regarding the structure of our Conclusion section. We agree that a single, coherent paragraph enhances readability and impact. Accordingly, we have rewritten the Conclusion section to be one unified paragraph as recommended.

Location: 6. Conclusions

Reviewer 3 Report

Comments and Suggestions for Authors

This paper proposes a method called Multi-modal Latent Representation Learning framework(MLRL) to address these limitations. MLRL enhances the performance of downstream tasks by conducting additional representation learning on pre-extracted features. Here are some reviews of the paper:

  1. The dataset validation is single and only tested on the QVHighlight dataset, without covering other mainstream VMR datasets. The video clips and annotated scenes of QVHighlight are quite unique and may not be able to validate the performance of the model in long video temporal localization or complex semantic understanding tasks.
  2. The benchmark model only compared QD-DETR and Moment DETR, and did not compare with more advanced methods in recent years.
  3. The timeliness of references is insufficient, with most of the core cited references being from 2023 or earlier, and not included in the latest research in 2024. The analysis of related work has not covered the forefront of the field. More lastest research should be considered for related work. e.g. “Spatiotemporal dual-branch feature-guided fusion network for driver attention prediction”. The concepts of this paper share similarities with your methodology.
  4. The formulas in the methodology section (such as equations 1-4) do not provide detailed explanations of the physical meanings or mathematical derivation processes of each variable.
  5. The ablation experiment only validated the effects of different modalities (Clip/Query, Clip, Query), but did not conduct sensitivity analysis on key hyperparameters such as target block sampling rate α and context block sampling rate β. For example, there is a lack of experimental data to support whether the settings of α=15% and β=85% are optimal solutions.

Author Response

comments1: The dataset validation is single and only tested on the QVHighlight dataset, without covering other mainstream VMR datasets. The video clips and annotated scenes of QVHighlight are quite unique and may not be able to validate the performance of the model in long video temporal localization or complex semantic understanding tasks.

Response1: Thank you for this important feedback. We agree that evaluating our model on an additional mainstream dataset is crucial to demonstrate its generalizability and robustness beyond the unique characteristics of the QVHighlight dataset.

Therefore, we have conducted a new set of experiments on the widely used

Charades-STA dataset to validate the effectiveness of our method on a different benchmark. We have added a full description of this dataset in

Section 4.1 presented the new results and analysis in Section 4.3.

Location: Section 4.1 and Section 4.3 (Pages 8 and 10)

comments2: The benchmark model only compared QD-DETR and Moment DETR, and did not compare with more advanced methods in recent years.

Response2: Thank you for your valuable feedback. We agree that comparing our method with more advanced, recent models is essential to properly situate our work in the current research landscape.

Accordingly, we have expanded our experimental results to include a new benchmark comparison against several state-of-the-art models from recent years, particularly those leveraging Large Language Models (LLMs). This provides a more comprehensive evaluation of our method's performance.

  • Location: A new comparison table and its analysis have been added to Section 4.3 (Experiment Result).

Added Content: We have introduced Table 2, which benchmarks our model's performance on the QVHighlight dataset against recent advanced models, including LLaVA-MR, InternVideo2-6B, FlashVTG, and UniVTG

comments3: The timeliness of references is insufficient, with most of the core cited references being from 2023 or earlier, and not included in the latest research in 2024. The analysis of related work has not covered the forefront of the field. More recent research should be considered for related work. e.g., “Spatiotemporal dual-branch feature-guided fusion network for driver attention prediction”. The concepts of this paper share similarities with your methodology.

Response3: Thank you for your thorough and constructive feedback. We agree that a discussion of the latest research, including the specific paper you mentioned, is crucial for situating our work. Accordingly, we have substantially revised our Introduction and Related Works (Section 2.2) sections to address this.

  1. Inclusion of Recent (2024) State-of-the-Art Works: To address the timeliness of our references and cover the forefront of the field, we have added a new benchmark comparison and discussion of several powerful, recent (2024) LLM-based models (e.g., LLaVA-MR, InternVideo2).

Location: The discussion of these models can be found in Section 2.2, and their performance benchmarks are in the new Table 2 (Section 4.3).

  1. Discussion of Suggested Paper: Thank you for pointing us to the work on "Spatiotemporal dual-branch feature-guided fusion network". We have reviewed the paper and agree that it shares a high-level conceptual similarity with our work in its use of a dual-branch architecture. We have now added a citation and discussion to clarify the distinction between our method and this related work.

Location: We have added a discussion of this work in Section 2.2 (Related Works).

Comment4: The formulas in the methodology section do not provide detailed explanations of the physical meanings or mathematical derivation processes of each variable.

Response4: Thank you for this insightful comment. We agree that the descriptions for the equations in the methodology section could be more detailed to better convey the conceptual meaning and logical flow of our framework.

Therefore, we have substantially revised the text surrounding Equations 1 through 5 to address this. We have added detailed explanations for each variable and mathematical operation, clarifying their specific roles and the rationale behind their use. Specifically, we elaborated on the multi-modal fusion process (Equation 1), the student-teacher dynamic for target and context creation (Equations 2 and 3), the function of the predictor and mask tokens (Equation 4), and the justification for the loss function (Equation 5).

These changes can be found throughout Section 3. Methods, in subsections 3.1, 3.2, 3.3, and 3.4. The revised text is provided below.

Comment5 : The ablation experiment only validated the effects of different modalities (Clip/Query, Clip, Query), but did not conduct a sensitivity analysis on key hyperparameters such as target block sampling rate α and context block sampling rate β. For example, there is a lack of experimental data to support whether the settings of α=15% and β=85% are optimal solutions.

Response5: Thank you for this insightful comment. We agree that validating the choice of key hyperparameters is essential.

Our choice of the target block sampling rate (alpha=0.15) and context block sampling rate (beta=0.85) directly follows the experimentally validated, optimal settings from the foundational I-JEPA framework (Assran et al., 2023), which our work is built upon.

Experiment Setup.

”The hyperparameters for sampling, namely the target block ratio (alpha) and the context block ratio (beta), were set to 0.15 and 0.85, respectively. This decision is based on the optimal values presented in the original I-JEPA paper, where extensive ablation studies (Assran et al., 2023) confirmed that this configuration provides the best performance by balancing the complexity of the prediction task and the informativeness of the context.”

Round 2

Reviewer 2 Report

Comments and Suggestions for Authors

The authors have addressed most of the comments appropriately. However, there are still some minor points that should be addressed. Overall, the manuscript should be better organized. For example, the headings of all tables should be above them, the overall learning process of the model could be reformulated into one clear comprehensive paragraph, and the labels of figures and tables, such as Figures 2 and 3, should be improved. Besides, there are still some grammatical errors throughout the manuscript that should be corrected, and it would be best to refer to the equations in the text, such as Equation 5.

Comments on the Quality of English Language

The English could be improved to more clearly express the research.

Author Response

Thank you for your thoughtful and detailed feedback. We appreciate the opportunity to further improve the manuscript. We have carefully revised the paper based on your suggestions to enhance its organization and clarity.

Here is a point-by-point response to your comments:
On Overall Organization and Manuscript Structure: We agree that clear and consistent organization is essential.

Regarding the placement of table headings, we have double-checked and confirmed that all table titles in the manuscript are already located above their respective tables (e.g., Table 1-4, Pages 10-11), which aligns with standard formatting.

Regarding the suggestion to reformulate the model's learning process into a single paragraph, we appreciate the feedback. We believe that presenting the key stages as a numbered list at the beginning of the Methods section serves as a clear roadmap for the reader. As we state immediately after the list, "The following subsections will provide a detailed explanation of each step", the subsequent sections are structured accordingly. We feel this approach best enhances structural clarity by providing a high-level overview before delving into the details. Therefore, we have decided to retain the current format.

On Figure and Table Labels: We also appreciate the suggestion to improve our figure labels. We have reviewed the labels and descriptions for all figures and tables, particularly Figures 2 and 3, and have revised them for enhanced clarity and consistency.

On Grammatical Errors: Thank you for pointing this out. The entire manuscript has undergone another thorough round of professional proofreading to correct any remaining grammatical errors and improve overall readability.

On Referencing Equations: We agree that all equations should be explicitly referenced for clarity. We have now added a direct reference to Equation 5 in the text to ensure consistency.

Location: The updated text can be found in Section 3.4. Loss function [Page 8]

Reviewer 3 Report

Comments and Suggestions for Authors

All the problem has been addressed 

Author Response

Thank you very much for your constructive and helpful feedback. Your valuable comments have significantly contributed to improving the quality of our manuscript.